# Vertical Slot Fishways: Incremental Knowledge to Define the Best Solution

**DOI:** 10.3390/biology12111431

**Published:** 2023-11-15

**Authors:** Paulo Branco, Ana Margarida Mascarenhas, Gonçalo Duarte, Filipe Romão, Ana Quaresma, Susana Dias Amaral, Maria Teresa Ferreira, António N. Pinheiro, José Maria Santos

**Affiliations:** 1Forest Research Centre, Associate Laboratory TERRA, School of Agriculture, University of Lisbon, 1349-017 Lisbon, Portugal; ana.mp.mascarenhas@gmail.com (A.M.M.); goncalofduarte@isa.ulisboa.pt (G.D.); samaral@isa.ulisboa.pt (S.D.A.); terferreira@isa.ulisboa.pt (M.T.F.); jmsantos@isa.ulisboa.pt (J.M.S.); 2Civil Engineering for Research and Innovation for Sustainability, Instituto Superior Técnico, University of Lisbon, 1049-001 Lisbon, Portugal; filipe.romao@tecnico.ulisboa.pt (F.R.); analopesquaresma@tecnico.ulisboa.pt (A.Q.); antonio.pinheiro@tecnico.ulisboa.pt (A.N.P.)

**Keywords:** river connectivity, vertical slot fishways, multiple slot fishways, river barriers, meta-analysis, freshwater fish

## Abstract

**Simple Summary:**

River barriers fragment longitudinal connectivity and limit the ability of several fish species to complete their life cycle. Vertical slot fishways (VFS) are the best technical fishway solution to enhance connectivity on artificial barriers. To (i) identify the variables affecting fish passage in VSF, and (ii) the best VSF for two freshwater fish species (*Luciobarbus bocagei* and *Squalius pyrenaicus*), we collected data from several fishway studies and applied Bayesian Generalized Mixed Models. Results show (i) that the main predictor for fish passage is fish size and (ii) from the tested configurations, the multiple slot fishway with an orifice was, overall, the best-performing fishway configuration for all fish sizes. Fishways, to be holistically effective, need to be designed with requirements for small fish.

**Abstract:**

River artificial fragmentation is arguably the most imperilling threat for freshwater-dependent fish species. Fish need to be able to freely move along river networks as not only spawning grounds but also refuge and feeding areas may be spatially and temporally separated. This incapacity of free displacement may result in genetic depletion of some populations, density reduction and even community changes, which may in turn affect how meta-community balances are regulated, potentially resulting in functional resilience reduction and ecosystem processes’ malfunction. Fishways are the most common and widely used method to improve connectivity for fish species. These structures allow fish to negotiate full barriers, thus reducing their connectivity impairment. Among all technical fishway types, vertical slot fishways (VSF) are considered to be the best solution, as they remain operational even with fluctuating water discharges and allow fish to negotiate each cross-wall at their desired depth. In the present study, we collected both published and original data on fish experiments within VSF, to address two questions, (1) What variables affect fish passage during experimental fishway studies? and (2) What is the best VSF configuration? We used Bayesian Generalized Mixed Models accounting for random effects of non-controlled factors, limiting inherent data dependencies, that may influence the model outcome. Results highlight that fish size, regardless of the species, is a good predictor of fishway negotiation success. Generally, multiple slot fishways with one orifice proved to be the best solution. Future work should be focused on small-sized fish to further improve the design of holistic fishways.

## 1. Introduction

Human societies have always been dependent on rivers. This has forced several pressures upon river systems, of which fragmentation by blockages in the form of dams or weirs is arguably the most pressing for fish [1,2]. These structures have always been used to regulate rivers and augment the ecosystem services provided by them. However, their placement fragments the natural longitudinal connectivity of rivers, impairing fish movements along river networks and imperilling fish biodiversity around the world [1,3].

In Europe, ca. 1.2 million barriers are projected to exist [4]. The increasing understanding and recognition of the threat that these structures pose to river systems-“Dams and water management” is the threat affecting more fish species in Europe [5]–led to the introduction of policies to reverse the river fragmentation process. The Water Framework Directive (2000/60/EC) clearly states that the re-establishment of longitudinal connectivity is paramount to attain the ultimate goal of Good Ecological Status (GES). Noteworthy is the recent European Biodiversity Strategy, which aims to reestablish connectivity along 25,000 km of European river networks by 2030. This is a valuable goal, but that has been considered underwhelming by a recent study [6] that stated and produced the groundwork, that 50,000 km would be a more effective attainable end goal. Nonetheless, a consensus exists on the impacts of river fragmentation and the merits of connectivity restoration. This restoration can be achieved in many ways, the most widespread of all being fishway construction, as it allows barriers to be kept in place and produce the services they were planned to provide [7]. The structural solutions for constructing fishpasses have been standard for quite some while, with few innovations being put forward. This lack of innovation stems from: (1) knowledge gaps about fish behaviour within fishways relative to less common or experimented fish species, and about fishway efficiency and effectiveness, so several studies have been focused on this [8,9]; and (2) to some extent, a disconnection between managers and scientists. Relying on a fixed set of rules may be detrimental to river connectivity, and several fishways are known to underperform [8] because rules were stipulated having a species or species group as targets, disregarding the rest of the fish species present at the river reach. This reasoning is easily synthesized by the “Field of Dreams Hypothesis” applied to fishways–“If we build it, they will come”. But, as stated before, one should build it properly–“If we build it properly, they will come”.

Among all the technical fishways, Vertical Slot Fishways (VSF) are considered to be one of the best [10]. They are characterized by having a continuous, top-to-bottom, opening (slot) in the cross-wall by which water flows between pools creating a streaming flow [11] and allowing fish species to negotiate the slot at their preferred depth across the water column. As such, it has been recommended as a holistic option to serve multiple fish species [12,13,14]. VSF also have the ability to remain operational, albeit with non-uniform flow profiles [15], under significant boundary conditions variations, which make them suited for rivers with high discharge fluctuations [16,17]. To negotiate the fishway, fish need to surpass the water velocity found at the slot, an ability limited by the size of the fish and by its ecomorphology. Thus, fishways may act as a selective pressure, hindering the movement of smaller fish or species with lower swimming capacity. Water velocities in the slots are dependent on the difference in the water level in two consecutive pools (head drop), which is ruled by the fishway slope, pool dimensions and flow regime [10]. Recently, a new type of VSF was developed [18,19] and a similar design was tested for its effectiveness under laboratory conditions [20,21]. This new development (Multiple Slot Fishway–MSF) incorporates two consecutive vertical slots between adjacent pools, reducing by 50% the head drop, decreasing water velocities and discharge needed to operate, making it less selective in terms of fish movements, and more cost-effective in terms of flow discharge.

To design more efficient and holistic fishways, a considerable experimental effort has been devoted to studying the effects on fish behaviour of specific alterations to the fishway design or operation. These studies are usually focused on simple and often single research questions. This multitude of stand-alone research articles gives us the opportunity to produce meta-analyses that extract more than the sum of isolated research-driven insights. Even marginal increases in data may improve our understanding of the problem. It may also allow us to further ascertain the impact of the experimentation procedure itself on the results, thus contributing to improving the experimental design of future studies.

Contrary to the deterministic overlook of the frequentist approach, Bayesian statistics represent a rigorous and coherent probability framework where powerful and flexible models can be established [22,23]. For this reason, the adoption of these methods is now widespread across multiple scientific disciplines, including ecology, and several statistical tools and techniques have been developed [23], including for regression models [22,24]. Generalised Linear Mixed Models (GLMM) are regression models using non-Gaussian distributions to model dependent variables via linear combinations of predictor variables (fixed effects) while considering levels of grouping variables (random effects) [25]. Fixed effects measure the overall effects of explanatory variables, while random effects can be used to estimate the amount of variation across grouping variables (e.g., species), or for nuisance variables (e.g., sampling hour) [26]. As such, this is an adequate statistical tool to work with data from multiple experiments.

In this work, we employed a Bayesian approach to answer two main questions: (1) What variables affect fish passage during experimental fishway studies?; and (2) What is the best VSF configuration? For this, we have collected a large body of published peer-reviewed literature on VSF experimental laboratory studies using wild fish conducted within the same fishway flume, with the same experimental design and by the same researchers. This data set comes from 5 recently published articles and from two unpublished tested configurations conducted at the same time and by the same team. This allows us to generate a data set that inherently controls for major non-test variables, such as flume dimensions, experimental protocol, flume slope, water quality, water temperature, flume operators and experiment team, variable retention and definition, fish sampling, fish transport, fish acclimation and maintenance protocols; and thus allow to focus on experimental variables. This significantly increases data and thus evolves from single comparisons of configurations or species performed in the stand-alone articles to more complex, overarching and far-reaching questions.

## 2. Material and Methods

### 2.1. Experimental Facility

The experimental facility has been fully described in previous publications e.g., [10,21,27]. It consists of an indoor full-scale model of a VSF (10 m long × 1 m wide × 1.2 m high) installed at the National Laboratory of Civil Engineering (LNEC), presenting mobile components (slots, deflectors) allowing to test different configurations. The structure base is made of steel, with 6 pools separated by five wood cross-walls (thickness: 22 mm) and featuring glass walls on both sides to allow continuous tracking of fish movements. The fishway slope was set at 8.5%, which is within the range of slopes found for this type of facility [17,28]. The fishway also incorporated (i) an upstream stilling chamber (1.85 m long × 1.00 m wide × 1.20 m high), which provided flow through a pump, measured by an electromagnetic flow-meter installed in a recirculation pipe, and (ii) a water storage tank (4.00 m long × 3.00 m wide × 4.00 m high) at its downstream end. To allow for prior fish acclimation to the flow, two mesh panels distancing 1.50 m apart, were placed at the most downstream tilting portion of the fishway, creating a 1.5 m^2^ acclimation area. Water quality parameters (temperature, conductivity and pH) during the experiments were daily monitored using a multiparametric probe (HANNA, HI 9812-5, Póvoa de Varzim, Portugal.

Six different configurations were tested (Table 1, Figure 1) based on the number of slots per pool—1 slot (standard VSF, 3 configurations) vs. 2 slots (multiple slot fishway, MSF, 3 configurations), slot orientation (aligned vs. alternating) and baffle position (central and lateral vs. only lateral) within the slots. Three configurations follow the VSF typology: (i) VSF with aligned slots featuring a central and lateral deflector (VSF_CD, corresponding to slot configuration C1–Romão et al., 2017) based on design 1 proposed by [29], (ii) VSF with aligned slots featuring a lateral deflector (VSF_LD–corresponding to slot configuration C2–[10]) based on design 11 proposed by [29] and (iii) VSF with alternating slots (VSF_ALT) and no deflectors. The other three configurations follow the MSF typology and were based on the Enature fishway concept [19]: (i) standard MSF (MSF); (ii) MSF with an orifice instead of the first slot (MSF_O), i.e., a combined solution that incorporated a vertical slot and a bottom orifice; and (iii) MSF with an orifice and notch in the first slot (MSF_ON). All configurations were tested with the same settings, namely (a) mean water depth in the pools (hm = 0.8 m), (b) slot width (b = 0.10 m), (c) pool length (L = 1.87 m), (d) pool width (B = 1.00 m) and (e) head drop between pools (ΔH = 0.16 m). Due to the twofold number of slots of MSF (2) when compared to VSF (1), the ΔH in the former is split into two, becoming approximately 0.08 per slot. This implies that, for the same hm, the MSF operates with a lower discharge (Q) compared to the VSF [27]. Data from VSF_CD, VSF_LD, MSF and MSF_O was retrieved from [10,20,21,27,30].

### 2.2. Fish Capture and Holding

The two selected species were chosen because they are representatives of two morpho-ecological guilds [31]. Iberian barbel (Luciobarbus bocagei, Steindachner, 1864;—hereafter barbel) represents large potamodromous fish and Iberian chub (Squalius pyrenaicus, Günther, 1868;—hereafter chub) represents small water column resident fish. Guilds tend to be considered a good way of representing groups of organisms that use the same resources [32].

Wild-adult barbel and chub were captured by wadable electrofishing (IG-200, Hans Grassl, Schönau am Königsee, Germany) in the Lizandro (barbel) and Lage (chub) Rivers. These are small coastal water courses (catchment area < 100 km^2^) that drain to the Atlantic Ocean. Capture methodology followed the CEN (2003) protocol and consisted of sampling in a single pass and throughout a 150-m long reach, the entire width of the river by walking slowly upstream in a zigzag pattern to ensure coverage of all habitats. Two sampling campaigns were performed: (i) spring (May to July), the reproductive migratory period when fish undergo upstream movements within rivers, and (ii) early fall (September–October), when shifts in home range associated with the search of feeding and refuge area occur [33]. To avoid potential bias on swimming performance from using individuals from different life-history stages, only adult individuals from both species—15–32 cm total length (TL) (barbel), 6–20 cm TL (chub)—were selected for the experiments. The sex of the individuals was not assessed as former studies have not found a relationship between this variable and swimming performance [34]. Fish were then transported in a 190 L fish transport box (Hans-Grassl, Schönau am Königssee, Germany) containing river water and featuring portable aerators (ELITE, Holm, Germany) to the Water Resources and Hydraulic Structures Unit of LNEC. Here they were placed for recovery (fishing and handling stress) and acclimation at ambient temperature and natural photoperiod in a 700 L filtered (Fluval Canister Filter FX5, turnover rate: 2300 L/h) and aerated acclimation tanks for 48h before the beginning of experiments [20,21]. Fish were not fed before the experiments [10], but food (Tetra Pond sticks) was provided thereafter daily to the tested fish, before being released in their natural habitats. Water quality in the acclimation tanks was monitored daily with a multiparametric probe (HANNA, HI 9812-5, Wensojit, RI, USA) for temperature, pH, and conductivity.

### 2.3. Fish Experiments

Experiments (75) employing a total of 225 barbels [mean ± standard deviation, total length (TL) = 19.9 ± 3.4 cm; total body mass = 78.6 ± 46.2 g] and 150 chubs [TL = 13.0 ± 2.1 cm; total body mass = 26.4 ± 15.3 g], were conducted in the spring and early fall of 2015 and 2016, in the week after fish sampling (see above for details). The sample size for each species was defined in a balanced way to both harmonize ethical issues while being able to answer scientific questions and gain relevant knowledge to improve species management and passability of these types of fishways, therefore following the principle of replacement, reduction, and refinement (3Rs), outlined in the European Directive 2010/63/EU, and transposed into Portuguese national legislation [35,36]. Overall, 75 trials were conducted: 20 barbel and 10 chub barbel trials in the spring and 25 barbel and 20 chub trials in early fall. Each trial was composed of a school of 5 adult fish (fish were previously selected to present a similar length to avoid any potential bias in the swimming performance) being the unit of analysis instead of each individual fish, as both species typically move that way to improve hydrodynamic efficiency [37]. Both species were tested separately, with each fish being tested only once to avoid any possible learning-based effects [38]. Before experimentation, each school was subjected to a 30-min acclimation period, enabling fish to adapt to the flume flow conditions. After that time, the upstream mesh panel delimiting the acclimation area was removed, and fish were able to volitionally move across the fishway for 90 min. Since both upstream and downstream movements were allowed, fish could successfully negotiate the fishway multiple times. The number of successes (dependent variable)—recorded by two independent observers aided by a video-recording camera (GoPro HERO5) positioned laterally to the fishway—was therefore considered as the sum of all upstream movements of every individual through the most upstream slot, thus effectively reaching the top of the fishway. Water quality (temperature, conductivity and pH) in the fishway was monitored continuously after each experiment using the same multiparametric probe (HI9812–5; HANNA Póvoa de Varzim, Portugal) as in the acclimation tanks. All experiments—75 in total, summing up 375 fish from both species—were performed by the same team, granting data consistency and thus reducing the confounding effects of experimentation. After experimentation, fish were then released into their natural environment.

### 2.4. Statistical Analysis

We performed two Bayesian General Linear Mixed Model (GLMM) analyses applying the Integrated Nested Laplace Approximation (INLA) analytical approach [39] via the INLA package [40] in R [41]. In both approaches, the negative binomial was used as the error distribution since the response variable was the number of successes in each experimental run. Exploratory analysis concerning outliers, data distribution and collinearity was performed following [42]. Variables with a Variance Inflation Factor (VIF) above 2.5, a threshold indicative of values that do not impose considerable collinearity [43], were sequentially excluded while factor variables data-wise unbalanced between levels were also excluded from the analysis (continuous variables present in Table 2 already reflect this selection process), see table S1 for excluded variables. Random variables may be selected if they affect variance in data, commonly termed nuisance variables [26,44]. As such, given that the two species used in the experiments present distinct adult body sizes and forms [2], different instream habitat preferences [45] and species-specific ecological traits we opted to consider this as a random factor variable. Concerning the first objective, to assess which experimental condition variables might affect the experimental outcome, we also included the fish-way design as a random factor variable since it may influence the number of successes in the experience. Using the same rationale, to determine the best fishway configuration (second approach) out of the 6 analyzed configurations, the factor variables that in the first approach come as significantly related to the response variable should then be included as random factors. No model selection was adopted due to the low number of fixed effect variables included in the models, and because model selection is controversial e.g., [46,47] and variable selection presents relevant drawbacks [22,48]. Variables were considered significantly related to the response variable if the value zero is not part of the 95% credible interval for their respective regression parameter [24]. Table 2 indicates and details the variables considered for both modelling procedures and their respective role in each procedure.

Model adequacy was assessed through graphics of residuals vs. fitted values and all covariates, including those removed in the exploratory analysis [22,24]. We assessed overdispersion by comparing the average squared Pearson residuals using the real data and data simulated from the model [22,24]. Since we are determining how frequent values from one set of residuals are higher than the other, an overall value around 0.5 indicates no dispersion issues while values closer to 0 and 1 reveal, respectively underdispersion and overdispersion [22,24]. Model validation, adequacy and visualization were achieved using the scripts made available in [22], the INLAutils package [49] and the INLAtools package [50].

## 3. Results

Overall, 55 experimental runs were taken in 2016 and 20 in 2015, while 45 occurred in the fall and 30 during the spring. Six different fishway configurations were tested, three belonging to the MSF typology with 40 experimental runs (MSF—20; MSF_O—10; MSF_ON—10) and three of the VSF typology with 35 experimental runs (VSF_ALT—5; VSF_CD—15; VSF_LD—15).

Concerning what affects fish passage during experiments (question one of this study), the exploratory analysis led to the selection of seven variables to be included in the model training, but given the values of the 95% credible interval of the regression parameter, only three (length_avg; Temp_PPP; Day_period_Morning) are considered to be affecting fish passage during the experimental trials (Table 3). All these variables present a positive effect on the response variable (the number of successes), with the highest effect being with the period of the day ($\bar{\beta }=0.389$) suggesting that a higher number of successes occurs in experimental runs taken during the morning period (Figure 2a). Other than this, a higher number of successes was also found to be related to fish schools with higher average fish lengths ($\bar{\beta }=0.288$; Figure 2b) and with higher water temperature during the experimental run ($\bar{\beta }=0.243$; Figure 2c). The full model presents a more parsimonious DIC (Deviance information criterion) and WAIC (Watanabe-Akaike information criterion) values and a fit dispersion value while graphical model validation does not reveal any indication of model misspecification (see Appendix A).

For the assessment of the best configuration (question two of the study), from the continuous variables detailed previously, coming from question one, only the average fish school length maintains a relevant effect ($\bar{\beta }=0.338$) when taking into consideration the fishway configuration (Table 4). Results indicate that fish schools with larger animals will tend to have a higher number of successes (Figure 3a). The other relevant variables were the fishway configuration for MSF_O and VSF_ALT, all revealing a positive relation with the dependent variable (Table 4). Overall, the MSF_O fishway configuration ($\bar{\beta }=0.961$) appears to have the highest effect on the number of successes in experimental trials, followed by the VSF_ALT configuration ($\bar{\beta }=0.886$) (Figure 3b,c). These are the only configurations that lead to a higher number of successes in fishway negotiation. As in the previous model, the model with all the variables is also more parsimonious than the null model, presents an adequate dispersion value and the graphical model validation has not revealed any indication of model misspecification (see Appendix A).

## 4. Discussion

Getting fish past the barriers is still the most important goal to reduce impacts created by artificial fragmentation [51]. This is no simple or easy task. Since the early works by Larinier [52,53,54] and Rajaratnam [29,55], this intent has been the subject of continuous research e.g., [56,57]. But, in a time where fishway researchers are understandably pushing for holistic solutions that can serve multiple species year-round [2,30,58], difficulties in experimental research are relegating research to simple near-reaching questions answerable with short, low replicate, and thus low animal use experiments. This, although serving a specific purpose, detracts from placing more general questions. This is the reasoning behind this study, where we set out to collect a controlled data set on vertical slot fishways experiments conducted with wild fish, to cast a wider net for more relevant questions and test the concept of marginal data gains incrementing significantly the response ability and data treatment possibilities.

Addressing question one, the results show that when fish experimentation within fishway flumes is planned correctly there is no significant impact of environmental variables (i.e., season, time of day and water conditions) on the fish passage of barbel and chub. In contrast, physical variables under test, like fishway configuration, tend to be the most impactful variables on fish performance within fishways. Furthermore, time of year, and year itself did not affect the results, meaning, as already put forward by Romão et al. [30], that fishways experiments can be conducted year-round and that experiments conducted in different years can be compared or their data pooled together. within reason. This result opens up a whole new world of possibilities for experimentation while reinforcing the idea that fishways should and can be operational and effective year-round [59]. By being able to compare experiments over and between years, researchers have the possibility to reduce experimental effort and, by extension, the number of wild fish used, which aligns with the 3R’s [35,36,60] recommendations [61] and the ethical guidelines of most countries and institutions, while increasing statistical power. Simply put, there is no significant need to constantly repeat control treatments when testing configuration variations of previously tested fishway solutions. 

Addressing question two, another important issue stood out in the results, even after the integration of fishway configurations as candidate variables: fish size is decisive for fishway negotiation of barbel and chub. But, because variation explained by species was removed from the analysis, we can say that irrespective of species or fish-way configuration, the larger the fish, the higher the probability of negotiation success. This means that, overall when looking at species with similar swimming types, a larger fish from a lower swimming ability species may have higher success than smaller fish from a species with a higher swimming performance. The fact that larger fish have a greater swimming ability is something long known [62,63,64], but the results of the present study give us more depth into the understanding of fish negotiating fishways. This result goes beyond those of previous studies where it is shown, for specific fishway configurations, that ecomorphological variability between species affects within fish-way behaviour and overall fishway effectiveness [2]. Of course, fish size does not relate exclusively to swimming ability it may have a scale effect in relation to fishway size and an effect on the advantage or disadvantage felt by fish navigating through turbulence fields with different size eddies [65] within fishways. Future studies need to focus on smaller fish, either that being earlier life stages of some species or small-sized fish species. If one of the contemporary goals of fishways is to serve all fish present at a given site and thus really contribute to proper ecosystem functioning, working on fishway efficiency enhancement for small fish is a crucial requirement [66]. If a small fish negotiates a fishway, a bigger healthy and motivated fish, not physically constrained by slot width (in the case of VSF) should also negotiate it.

When looking at the results as a whole, two fishway configurations outperform all the others, VSF_ALT (vertical slot fishway with alternating slots) and MSF_O (multiple slot fishway with an orifice). However, the latter is supported by a larger number of trials, allowing the model outputs to probably be more representative of how a fishway would be effective in allowing fish to negotiate it in a real-world context. These multiple slot solutions are a recent state-of-the-art innovation [19] that have several advantages: (i) they are more water cost-effective, being a relevant solution to allow system longitudinal continuity in water scarcity regions, like Mediterranean river systems, where the operation with lower flow may allow continuous operation [21]–this advantage can be even more relevant in the context of global changes; (ii) by operating with a lower discharge and (iii) by reducing the head drop, flow velocities at the slots and within pool turbulence, which benefits both small-sized species and smaller fish, something proved herein to be a touchstone for holistic fishway effectiveness. Nevertheless, these structures also present caveats that managers should account for when implementing them at river barriers: (i) as with all pool-type fishways, care must be taken to allow for a uniform flow regime, a caveat partially controlled by the inherent water cost-effectiveness of this fishway type [10]; (ii) there is added risk of clogging due to water transported debris, or debris falling from overhanging canopy, so managers should enforce a regular maintenance routine. Clogging seriously affects fishway operation, promoting non-uniformity along the fishway [15,51], and may, without regular maintenance, completely block the fishway making it nonoperational [10,67].

## 5. Conclusions

This work has demonstrated how well-designed experiments offset the effect of non-test variables and prove how a fish passage is mostly affected by the variables under test when co-variables are controlled for. It also reinforced the fact that fish will negotiate fishways throughout the year and that multiyear comparisons of different configurations are legitimate. Moreover, fish size is a strong predictor of fishway negotiation success, regardless of the species. The multiple slot fishway with an orifice (MSF_O) was shown to be the most effective and holistic configuration. All these results also demonstrate that incremental knowledge, by collecting published data from stand-alone articles, is a way of adding value to published literature, allowing research to depart from simple questions and move towards general questions. We can thus use incremental knowledge to generalize specific discoveries and broaden their relevance, benefitting species conservation and improving systems functioning. This approach should now be extended to additional datasets, and focus on additional species while trying to pursue different questions. The general incremental knowledge framework should also target additional longitudinal connectivity conundrums such as fish negotiation of small instream obstacles and weirs e.g., [3,4,68]. As relevant as they are, laboratory-controlled studies should, as much as possible, be complemented by field studies. The approach followed herein can be used to determine the best possible solutions, but these should be tested under real-world conditions in the field to allow for proof-of-concept before generalized implementation.

### Ethical Note

All the procedures that involved fish handling, capture, transportation, housing and experimentation were performed in compliance with European [60] and Portuguese legislation [35]. Sampling, transportation and housing permissions according to the outlined methodology were issued by the Institute for Nature Conservation and Forests (ICNF). Experiments were conducted according to the guidelines of the Protection of Animal Use for Experimental and Scientific Work of the Department for Health and Animal Protection (Direção Geral de Alimentação e Veterinária, see https://www.dgav.pt/animais/conteudo/animais-para-fins-cientificos/bem-estar-animal/, accessed 13 July 2023), which authorised animal experiments to be performed in the experimental facility. JMS which holds FELASA level C certification (www.felasa.eu, accessed on 1 Septembre 2015) is authorized to carry out, design and coordinate projects involving animal experimentation. No fish were sacrificed for this study.

## Figures and Tables

**Figure 1 biology-12-01431-f001:**
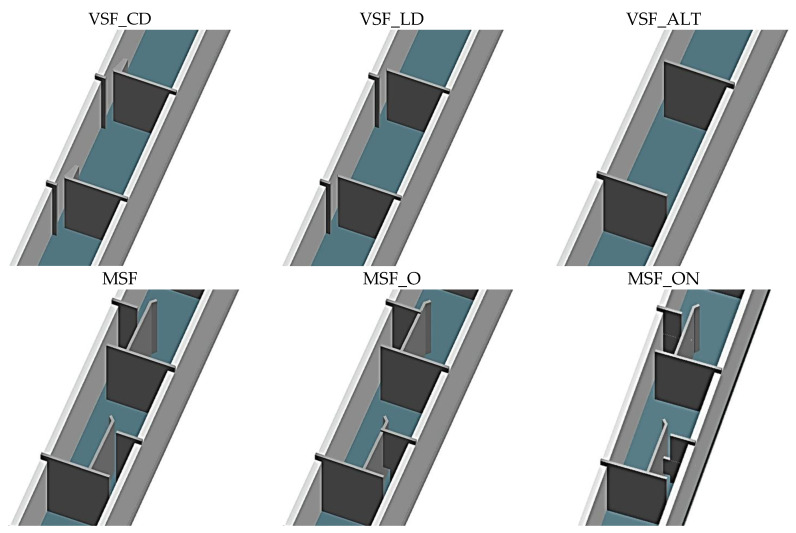
Diagram of the fishway flume used in the experiments (see Table 1 for details). Top view schematics of examples of vertical slot fishways (VSF) and Multiple slot fishways (MSF).

**Figure 2 biology-12-01431-f002:**
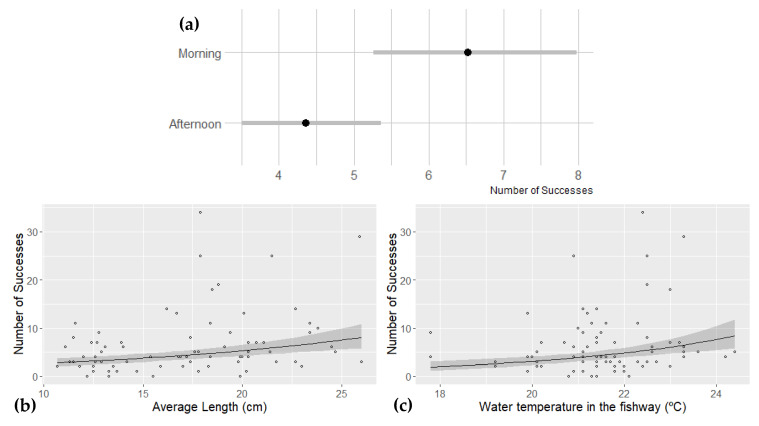
Predicted response in terms of number of successes in relation to (**a**) the period of the day (black dots represent the average predicted value and the grey lines the 95% credible interval), (**b**) the average fish school length; and (**c**) the water temperature in the fishway. In (**b**,**c**) data points are represented by black dots and the grey-shaded area represents the 95% credible interval. Species used in this work: *Luciobarbus bocagei*; *Squalius pyrenaicus*. Species was a random factor in the modelling procedure.

**Figure 3 biology-12-01431-f003:**
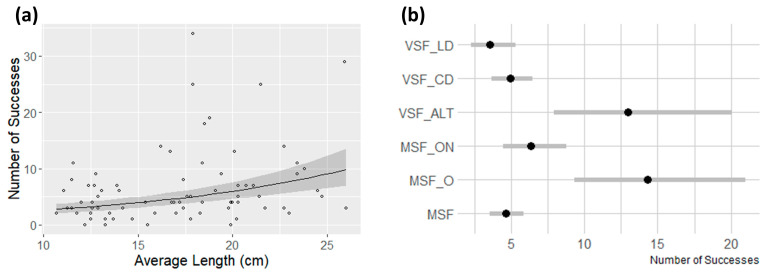
Predicted response in terms of number of successes in relation to (**a**) the average fish school length, (**b**) the fishway configuration (black dots represent the average predicted value and the grey lines the 95% credible interval) and (**c**) the combination of the fishway configuration with the average fish school length. In (**a**,**c**) data points are represented by black dots and the grey-shaded area represents the 95% credible interval. VSF_CD—Vertical slot fishway (VSF) with central and lateral deflectors. VSF_LD—VSF with lateral deflector. VSF_ALT—VSF with alternating slots. MSF—Multiple slot fishway (MSF). MSF_O—MAF with an orifice instead of the first slot. MSF_ON—MSF with orifice and notch in the first slot.

**Table 1 biology-12-01431-t001:** Summary of the Vertical Slot Fishway’s (VSF) configurations used for this study. MSF—Multiple Slot Fishway; Season—Season(s) in which the experiments were conducted. Q (L/s).

Configuration Code	Configuration Description	Flow Discharge (L/s)	Season
VSF_CD	VSF with central and lateral deflector	110	Spring and Fall
VSF_LD	VSF with lateral deflector	81	Spring and Fall
VSF_ALT	VSF with alternating slots	88	Spring
MSF	MSF	56	Spring and Fall
MSF_O	MSF with an orifice instead of the first slot	36	Spring
MSF_ON	MSF with orifice and notch in the first slot	33	Fall

**Table 2 biology-12-01431-t002:** Details of the variables considered for both models and their respective roles in each model.

Type of Variable	Variable	Acronym	Description	Unit/Levels	Role in Modelling Procedure 1	Role in Modelling Procedure 2
Ordinal	Number of successes	Success_num	Number of times a fish overcomes the entire experimental fishway	—	Response variable	Response variable
Factor	Species name	Species	Identification of the species from which the fish used in the experimental run belong to	Iberian barbel and Southern Iberian Chub	Random effect	Random effect
Factor	Seasons	Season	Identification of the season in which the experiment took place	Spring and Fall	Fixed effect	—
Factor	Period of the day	Day_period	Identification of the period of the day in which the experiment occurred	Morning (from 7 to 12 am) and afternoon (from 12 to 6 pm)	Fixed effect	Random effect
Continuous	Average length of fish school	Length_avg	Average total length considering the five fish present in each fish school	cm	Fixed effect	Fixed effect
Continuous	Water temperature in the fishway	Temp_PPP	Water temperature in the fishway measured before the experiment	°C	Fixed effect	Fixed effect
Continuous	Water conductivity in the fishway	Conduct_PPP	Water conductivity in the fishway measured before the experiment	µS cm^−1^	Fixed effect	—
Continuous	Water conductivity during acclimatation	Conduct_aclim	Water conductivity in the aclimatation tanks	µS cm^−1^	Fixed effect	—
Continuous	Water pH during acclimatation	pH_aclim	Water pH in the aclimatation tanks	—	Fixed effect	—
Factor	Fishway configuration	Fishway_config	Configuration of the fishway used for each experimental run	MSF — MSF_O — MSF_ON —VSF_ALT —VSF_CD — VSF_LD —	Random effect	Fixed effect

**Table 3 biology-12-01431-t003:** Deviance information criterion (DIC), Watanabe–Akaike information criterion (WAIC) and dispersion parameters for both the null model and the full model concerning the first question (What variables affect fish passage during experimental fishway studies?). Values of the standardized regression parameter for each variable of the full model are also provided for the average (mean), the standard deviation (sd), the lower value of the 95% credible interval (Lower_CI) and the higher value of the 95% credible interval (Upper_CI). Variables significantly related to the response variable are underlined. Species was considered as a random factor variable in the model.

**Null Model**	**Model** **Parameters**	**DIC**	**WAIC**	**Dispersion**	
		440.95	441.65	0.91	
**Full Model**	**Model** **parameters**	**DIC**	**WAIC**	**Dispersion**	
		428.68	429.31	0.5	
	**Explanatory** **Variables**	**Regression Parameter**			
		**mean**	**sd**	**Lower_CI**	**Upper_CI**
	length_avg	0.288	0.097	0.098	0.479
	Temp_PPP	0.243	0.110	0.025	0.460
	Conduct_PPP	0.078	0.111	−0.140	0.298
	Conduct_aclim	−0.003	0.128	−0.253	0.248
	pH_aclim	−0.224	0.131	−0.481	0.034
	Season (Spring)	0.171	0.205	−0.233	0.573
	Day_period (Morning)	0.389	0.187	0.020	0.757

**Table 4 biology-12-01431-t004:** Deviance information criterion (DIC), Watanabe–Akaike information criterion (WAIC) and dispersion parameters for both the null model and the full model concerning the second question (What is the best vertical slot fishway configuration?). Values of the standardized regression parameter for each variable of the full model are also provided for the average (mean), the standard deviation (sd), the lower value of the 95% credible interval (Lower_CI) and the higher value of the 95% credible interval (Upper_CI). Variables significanty related with the response variable are underlined.

**Null Model**	**Model** **Parameters**	**DIC**	**WAIC**	**Dispersion**	
		440.95	441.65	0.91	
**Full Model**	**Model** **Parameters**	**DIC**	**WAIC**	**Dispersion**	
		425.42	427.3	0.532	
	**Explanatory** **Variables**	**Regression Parameter**			
		**mean**	**sd**	**Lower_CI**	**Upper_CI**
	length_avg.std	0.338	0.094	0.154	0.522
	Temp_PPP.std	0.036	0.113	−0.186	0.258
	Fishway_config (MSF_O)	0.961	0.304	0.365	1.559
	Fishway_config (MSF_ON)	0.284	0.304	−0.313	0.884
	Fishway_config (VSF_ALT)	0.886	0.376	0.148	1.629
	Fishway_config (VSF_CD)	0.102	0.282	−0.451	0.658
	Fishway_config (VSF_LD)	−0.081	0.277	−0.623	0.466

## Data Availability

No new data were created in this study.

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
