# Peer review of "Vertical Slot Fishways: Incremental Knowledge to Define the Best Solution"

_biology, 2023, doi:10.3390/biology12111431_

Round 1
Reviewer 1 Report
Comments and Suggestions for Authors
The authors present a meta-analysis on previously published experimental studies on vertical-slot fishway. The authors make excellent use of existing data to expand on simple scientific inquiry to more generalizable observations / data needs. I laud this type of exercise as it maximizes the insights gained through iterative studies on similar infrastructure and fish species. The analysis revealed, almost unsurprisingly, that fish length, time of day, and temperature had the most influence on passage success. Similarity, the examination of the best configuration also determined that fish length was important and that the MSF_O and VSF_ALT fishway types resulted in the most passage. Overall, the manuscript is well written and the statistical methods are sound. However, I found the key objectives of the study to be poorly formulated and conclusions to be overstated.
In general terms, the objectives sought to find 1) what factors affect fish behavior in a VSF and 2) what VSF configuration is best. With respect to 1), the authors need to be more explicit when mentioning “fish behavior” as this is a more nebulous variable, when it is clear from the analysis that the authors are only examining passage success. The study of fish behavior within a fishway and in relation to hydraulic and other environmental conditions eludes to much more than simply passage. With respect to 2), the authors should acknowledge that the subset of configurations is rather limited to 6 distinct configurations. I was expecting insights regarding fishway size, slope, slot width, orifice size, flow rate, and location, etc. But in realty the analysis only considered 6 configurations under static flow conditions.
While the methods and results sections were clear, I found the discussion section and its recommendations to be unsupported by the study. For example, L 379-382 state that fishway experiments can be conducted at any time of year and that data across years can be pooled. This all assumes that all other confounding variables are identical, which is unlikely. I find this statement to be too broad a generalization that could be dangerous in practice. To assume that fish behavior in a fishway is consistent across time is simply not supported by the literature. Another example would be in regards to L 391-408 which suggests fish size impacts passage success purely due to swimming performance. This statement neglects the effects of scale on access of slower moving water in boundary layers or destabilizing effect of turbulence of similar scale of the fish. Finally, the discussion references to fish generally while the study only considered two species.
Based on these concerns, I cannot recommend the manuscript for publication at this time. I feel these issues can be addressed within a revision and will be a solid contribution to the journal.
Comments on the Quality of English LanguageOnly minor edits are required.
Author Response
All answers are in the word file

Reviewer 2 Report
Comments and Suggestions for Authors
1. The abstract of the article is too long and should be modified appropriately to make the language more concise.
2. The introduction part of the article is too long and needs to be revised. it should give a systematic summary of the previous research and at the same time indicate the strengths as well as the innovations of the content of the work in this paper.
3. Why the fish used in the experiment are barbels and chubs, give the basis.
4. What is the rationale for the VSF setup in chapter 2 of the paper, and whether there are other more reasonable setups?
5. The discussion in Chapter 4 should be based on the experimental data collected and analyzed, and the results of the experiments should be discussed accordingly.
6. The article lacks a conclusion section.
Author Response
All answers are in the word file

Reviewer 3 Report
Comments and Suggestions for Authors
Review for the paper “Vertical slot fishways: incremental knowledge to define the best solution” by Paulo Branco, Ana Margarida Mascarenhas, Gonçalo Duarte, Filipe Romão, Ana Lopes Quaresma, Susana Dias Amaral, Maria Teresa Ferreira, António N Pinheiro, José Maria Santos submitted to "Biology".
Migration is a critical aspect of animal behavior, as it maximizes survival and reproductive success by moving between essential habitats, such as feeding and reproductive areas. Unfortunately, anthropogenic disturbances have been progressively fragmenting habitats, consequently inhibiting animal migrations. Over the past century, populations of many migratory species have declined dramatically. Freshwater systems have undergone significant decline in terms of habitat connectivity. Numerous waterways have been obstructed or embanked, leading to a highly regulated water flow, poorly developed riparian zones and increased sediment loads. Furthermore, these waterways are experiencing nutrient transport loss, eutrophication, and/or pollution, ultimately causing deficits in crucial habitats required for the growth and spawning of fish. These barriers greatly increase the extirpation risk of freshwater fish species.
The authors utilized experimental data to simulate fish migrations through artificial fishways and determine an optimal fishway design. The study aimed to identify factors that facilitate successive passage of two freshwater species, the Iberian barbel and Iberian chub. Results showed that fish size was the most dependable predictor of fishway negotiation success irrespective of species. The study also found that fishways featuring multiple slots and one orifice were the most effective design. These findings may have significant implications for fish conservation in Europe. The paper is well-written; however, some issues must be taken into account before the paper can be accepted.
Recommendations:
1) A concise summary is required for the “Biology” journal, so the authors ought to include one.
2) The abstract should be condensed to 200 words in line with the Instructions for Authors. The information in Lines 9-20 is commonly known, and can therefore be shortened.
3) The statements in Lines 198-208 require appropriate citations to support them.
4) Lines 217-219. The authors should revise the text to clarify their data representation. For instance, they should present their data as “total length (TL), mean ± standard deviation = 19.9 ± 3.4 cm” instead of “mean ± total length (TL) =19.9 ± 3.4 cm”. Additionally, the authors should specify whether they used standard deviation (SD) or standard error (SE).
5) L 298-303. This information is repetitive to the text above and seems redundant.
6) L 304. As only section 3.1 occurs in the text, this sub-division makes no sense.
7) The authors' use of the term "CI" as an acronym for "credible interval" is uncommon, as this acronym is generally used to represent "confidence interval".
8) L 310-314. The authors found positive and significant effects of daytime and water temperature on fish behavior when passing a fishway but these findings were not discussed at all. The authors should update the discussion and provide explanations for their findings along with comparisons with results from similar studies.
9) Supplementary material was cited throughout the text, yet not made available.
10) The authors should adhere to the MDPI citation and reference style for the text.
Specific remarks.
L 15. Consider replacing “ecosystems processes' malfunction” with “ecosystem processes' malfunction”
L 42. Consider replacing “the World War II” with “World War II”
L 43. Consider replacing “which lead” with “which led”
L 47. Consider replacing “structures represent to river systems” with “structures pose to river systems”
L 49. Consider replacing “lead to the introduction” with “led to the introduction”
L 77. Consider replacing “top to bottom” with “top-to-bottom”
L 107. Consider replacing “contributing to improve” with “contributing to improving”
L 129. Consider replacing “This allows us to have a data set” with “This allows us to generate a data set”
L 133. Consider replacing “thus evolve from” with “thus evolves from”
L 190. Consider replacing “was not determined into account” with “was not taken into account”
L 268. Consider replacing “distinct adult body size and form” with “distinct adult body sizes and forms”
L 306. Consider replacing “analysis lead to the selection” with “analysis led to the selection”
L 323, 324, 351. Consider replacing “value of 95% the” with “value of the 95%”
L 329, 355. Consider replacing “Predicted responses number of successes” with “Predicted responses for number of successes”
L 366. Consider replacing “multiples species” with “multiple species”
Comments on the Quality of English LanguageMinor revision.
Author Response
All answers are in the word file

Round 2
Reviewer 2 Report
Comments and Suggestions for Authors
The author has modified it as requested